# DSBRouter: End-to-end Global Routing via Diffusion Schrödinger Bridge

**Liangliang Shi** [*1]  **Shenhui Zhang** [*2]  **Xingbo Du** [2]  **Nianzu Yang** [2]  **Junchi Yan** [†2]

## Abstract

Global routing (GR) is a fundamental task in modern chip design and various learning techniques have been devised. However, a persistent challenge is the inherent lack of a mechanism to guarantee the routing connectivity in network's prediction results, necessitating post-processing search or reinforcement learning (RL) to enforce the connectivity. In this paper, we propose a neural GR solver called DSBRouter, leveraging the Diffusion Schrödinger Bridge (DSB) model for GR. During training, unlike previous works that learn the mapping from noise to routes, we establish a bridge between the initial pins and the routing via DSB, which learns the forward and backward mapping between them. For inference, based on the evaluation metric (e.g. low overflow), we further introduce a sampling scheme with evaluation-based guidance to enhance the routing predictions. Note that DSBRouter is an end-to-end model that does not require a post-step to ensure connectivity. Empirical results show that it achieves SOTA performance on the overflow reduction in ISPD98 and part of ISPD07. In some cases, DSBRouter can even generate routes with zero overflow.

## 1. Introduction

Global routing (GR) (McMurchie et al., 1995; Liao et al., 2020; Cheng et al., 2022) has emerged as one of the most intricate and time-consuming phases in the modern design flow of Very Large Scale Integration (VLSI) (Kramer & Van Leeuwen, 1984), distinct from other stages such as logic synthesis (Neto et al., 2021), floorplannning (Li et al.,

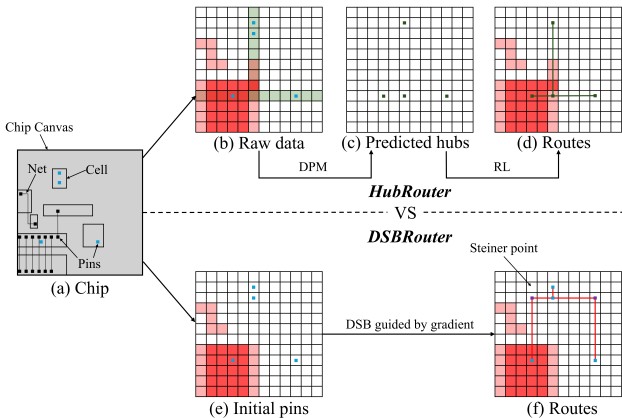

*Figure 1.* Global routing in chip design via different methods. (a) a real chip canvas from ISPD07. (b) The raw data is fed into DPM in HubRouter. The red stripe represents the congestion in the existing chip, while the green stripe is the stripe mask. (c) predicted hubs. (d) generated routes by RL or other post-processing algorithms. (e) Input pins of DSBRouter. (f) generated routes by DSBRouter.

2022a), placement (Hao et al., 2021; Shi et al., 2023c), etc. With VLSI netlists containing millions or billions of nets, global routers must interconnect pins while minimizing wirelength and avoiding overflow in a limited area. However, even in the simplified '2-pin' scenario that connects each net with only two pins under specific constraints, has proven to be NP-complete (Paulus et al., 2021).

Traditional works (Cho et al., 2007; Kastner et al., 2002) often rely on heuristics to solve greedily. However, the diversity and scale in the modern chip design industry, introduce new challenges for classical algorithms, requiring continuous updates and improvements by human experts. To reduce reliance on manual efforts and enhance overall design automation and quality, learning-based methods have been introduced, yet these methods suffer from notable limitations as shown in Tab. 1. For instance, approaches like (Liao et al., 2020) leverage deep reinforcement learning (RL) to obtain solutions but require substantial time for route generation. Others, e.g. HubRouter (Du et al., 2023) produce solutions directly via generative methods (Li et al., 2022b; Yan et al., 2018; Li et al., 2021). However, the generated results often rely on second-stage heuristics or RL techniques to ensure global connectivity. This raises a natural question: Can we directly generate high-quality, connected routes in

*Equal contribution †Correspondence author [1]Shanghai Institute for Mathematics and Interdisciplinary Sciences, Shanghai, China [2]School of Artificial Intelligence and School of Computer Science, Shanghai Jiao Tong University, China. Correspondence to: Junchi Yan <yanjunchi@sjtu.edu.cn>. This work is supported by NSFC (92370201, 62222607). Code available at https://github.com/Thinklab-SJTU/EDA-AI.

an End-to-End manner without second-phase corrections? This property is algorithmically nice as it aligns the training and inference without additional steps in inference.

In this paper, we propose DSBRouter, which utilizes the Diffusion Schrödinger Bridge (DSB) model (De Bortoli et al., 2021) for route generation. Unlike previous generative models (Du et al., 2023) that treat initial pins as conditions and learn a mapping from noise to routes, the proposed DSBRouter learns both the forward and backward mappings between initial pins and routing results, enhancing performance through alternating parameter learning. During the inference process, to better align with the final evaluation metrics (e.g., low overflow) for the routing task, we further introduce an evaluation-based guidance technique that improves the accuracy of the generated results. Extensive experiments show that DSBRouter significantly improves overflow and achieves state-of-the-art performance on public benchmarks. This paper contributes as follows:

1) To our best knowledge, we are the first to successfully introduce DSB (De Bortoli et al., 2021) to the global routing problem while incorporating the alignment of data, and the devised DSBRouter establishes the forward and backward mappings between the initial pins and the routing.

2) We show how to effectively integrate optimization evaluation-based guidance into the DSB model, with direct gradient feedback from the instance-wise objective score, enabling it to optimize while ensuring general feasibility (i.e. connectivity in this paper).

3) It achieves strong performance without post-processing techniques for the feasibility of final results. In particular, DSBRouter achieves SOTA performance and, in some datasets, even achieves an overflow reduction of 100%.

## 2. Related Work

### 2.1. Global Routing

**The Task of Global Routing.** Given the intricate nature of VLSI routing dilemmas, the circuit layout, such as Fig. 1a, is divided into rectangular regions termed global cells (Cho et al., 2007). The challenge of global routing can be conceptualized as a grid graph $G(V, E)$, where each GCell is depicted as a vertex ($v \in V$), and neighboring GCells are linked by an edge ($e \in E$) symbolizing their shared boundary. Chip designs commonly incorporate two or more metal layers for routing purposes. Each metal layer is assigned to either a horizontal or vertical orientation, and their mapping onto a two-dimensional grid graph is illustrated in Fig. 1b. The global router will designate a cluster of interconnected GCells, connected by multiple edges, to each net as its routing outcome to link all pins, typically forming a Rectilinear Steiner Tree (RST) (Chu & Wong, 2005). The principles

of Hanan grid (Hanan, 1966) and escape graph (Ganley & Cohoon, 1994) are frequently utilized to formulate the shortest RSMT while circumventing obstacles (Liu et al., 2012), considering the intersection points within these graphs as potential locations for Steiner points.

**Traditional Global Router.** Global routing is actually a combinatorial problem and can be formulated as a 0-1 integer linear programming problem thus one can solve it with a general solver. Traditional routing algorithms commonly split global routing into two primary stages to manage congestion: Steiner topology generation and rip-up and reroute (RRR). The former employs the FLUTE algorithm (Chu & Wong, 2005), utilizing lookup tables to create Steiner trees with minimal wirelength for each net. However, FLUTE does not consider congestion. During this phase, most routers use edge shifting to alleviate congestion by moving edges out of congested regions (Chu & Wong, 2005), while Nthu-Route 2.0 (Chang et al., 2008) introduces a novel history-based cost function that records and analyzes previous routing congestion, dynamically adjusting the routing cost to enhance overall routing quality and efficiency. To resolve congestion within the RSTs, traditional routers employ RRR, iteratively removing initially routed nets in congested zones and utilizing maze routing to optimize wirelength and congestion simultaneously. This process becomes significantly time-consuming as chip design complexity and scale increase. Thus, accelerating congestion resolution through deep learning-based methods can improve the overall performance of global routing algorithms.

**Learning-based Router.** Various works have explored the feasibility and benefits of optimizing wire length, as well as the effectiveness of applying neural networks to global routing. These studies include generating pin-connection orders (Liao et al., 2020), segments (Cheng et al., 2022), or customized hub points for rectilinear Steiner trees (RST) (Du et al., 2023). However, the primary challenges in practical global routing lie in managing the complexity of large-scale nets and preventing overflow when routing resources are limited. In such cases, detours play a critical role in mitigating congestion, since the shortest RST—such as the one shown in Fig.1 produced by HubRouter (Du et al., 2023)—may not be feasible in practice. The chip layout can be analogized to an image, where each pixel signifies a tile in global routing. Images of varying channels denote pin locations and grid edge capacities. The resultant points can also be represented as a binary image.

### 2.2. Diffusion Schrödinger Bridges and Applications

Given two marginal distributions and a reference stochastic process between them, the Schrödinger Bridge (SB) (Schrödinger, 1932) aims to find a process that minimizes the Kullback-Leibler divergence relative to the refer-

*Table 1.* Characteristics of global routing approaches. Classical and RL-based methods are sensitive to pin scale, while existing generative based methods typically decompose the task into a two-stage point generation and connection pipeline. DSBRouter, in this paper, only one model is needed to ensure end-to-end generation from large-scale pins to routes, while also considering the quality and connectivity of the generated routes, without any post-processing.

| Model | Type | Multi-pin | Connectivity | Scalability | End-to-End |
|---|---|---|---|---|---|
| DRL (Liao et al., 2020) | RL | ✗ | ✓ | ✗ | ✓ |
| PRNet (Cheng et al., 2022) | Generative | ✓ | ✗ | △* | ✓ |
| HubRouter (Du et al., 2023) | Generative + RL | ✓ | ✓ | ✓ | ✗ |
| NeuralSteiner (Liu et al., 2024) | ML + classical post-processing | ✓ | ✓ | ✓ | ✗ |
| DSBRouter (Ours) | Generative | ✓ | ✓ | ✓ | ✓ |

△* Scalable for one-shot generation, but not scalable for post-processing.

ence process in path spaces, which is a generalization of entropic optimal transport (Shi et al., 2023a; 2024b). Although a closed-form solution for the SB problem is generally not available, it can be approached numerically via Iterative Proportional Fitting (IPF) (Fortet, 1940) or Iterative Markovian Fitting (IMF) (Shi et al., 2023b). (De Bortoli et al., 2021) propose the Diffusion Schrödinger Bridge (DSB), viewed as a numerical approximation of IPF. The first iteration of DSB recovers the score-based generative model (SGM) introduced by Song et al. (2021b). With further iterations, DSB parametrizes not only the backward process but also the forward process. Subsequent studies (Shi et al., 2023b; Peluchetti, 2023) have built upon DSB, introducing refinements that further improve its theoretical foundations and practical applicability. In particular, Tang et al. (2024) proposes a theoretical simplification of DSB that integrates SGM as the initialization for DSB, accelerating its convergence. However, existing DSB algorithms generally fail to utilize the aligned data (Somnath et al., 2023).

Shi et al. (2023b) investigates the performance of DSB in various application scenarios, including generative modeling (De Bortoli et al., 2021), optimal transport (Villani et al., 2009; Peyré et al., 2019), and high-dimensional data transition (Chen, 2023; Hoogeboom et al., 2023). Unlike optimal transport (Shi et al., 2024c;a; 2025a), focusing on optimizing the transport path between two distributions, and unconstrained high-dimensional data transition problems (e.g., converting images of cats to those of dogs), our study addresses the Global Routing problem. It involves generating high-quality connected routing image distributions from the original circuit image distribution under constraints such as geometric and resource limitations. It represents a high-dimensional data transition problem with optimization constraints. The trained DSB can directly perform transitions between seen distributions. However, the Global Routing problem, with the incorporation of geometric and resource constraints, also requires transition on unseen circuit distributions, generating high-quality connected routes, which is undoubtedly a significant challenge. In this paper, a Router based on DSB and an objective-oriented gradient feedback optimization (i.e., DSBRouter) is proposed to address this challenge.

## 3. Methodology

In this section, we present our proposed DSBRouter, starting with an approach overview.

### 3.1. Preliminaries and Approach Overview

**Overflow (OF).** Let edge $e(u, v)$ represent the boundary between GCell $u$ and GCell $v$. The capacity $c(u, v)$ is the available routing resource on edge $e$, while the demand $d(u, v)$ is the number of routes that traverse it. The resource $r(u, v)$, which can actually be utilized for routing is defined:

$$r(u, v) = c(u, v) - d(u, v). \qquad (1)$$

When $r(u, v) < 0$, overflow occurs. Global routing results with excessive overflow may require a time-consuming rip-up and reroute process (Chen & Chang, 2009) or even placement adjustments to ensure successful routing. Hence, in addition to finding the shortest paths for each net, the global router must also aim to minimize overflow occurrences. To reduce OF in the generated routes. DSBRouter proposes an OF-oriented objective score function to perform sampling with guidance in the inference stage, which will be introduced in Sec. 3.3. The overview of DSBRouter is illustrated in Fig. 2. As an End-to-End learning model, DSBRouter takes the initial pins with congestion heatmap as inputs and directly outputs a route with strong connectivity. During training, defined in dsb as the backward process from pins to routes, we provide the model with distribution $X_N$ and train it to produce distribution $X_0$. The transition of $X_N$ to $X_0$ also relies on the noiser scheduler introduced in Tang et al. (2024), besides the backbone neural network of DSBRouter. In the inference stage, we introduce an RSMT-based approach to output the Steiner map $\mathbf{M}_s$(shown in the right part of Fig. 2) based on the predicted routing results $S_0$ at any given timestamp. Then the suboptimal intermediate expected route $S_1$ is generated given $\mathbf{M}_s$ through algorithm 1. The objective score function $\mathbf{g}^*$ is then employed to compute the gradient on any corner or Steiner pins (Liu et al., 2024; Du et al., 2023) in $S_1$ to perform the evaluation-based guidance, as detailed in algorithm 2. After several rounds of optimization guidance by above-mentioned steps, the final generated route is refined, preserving connectivity while

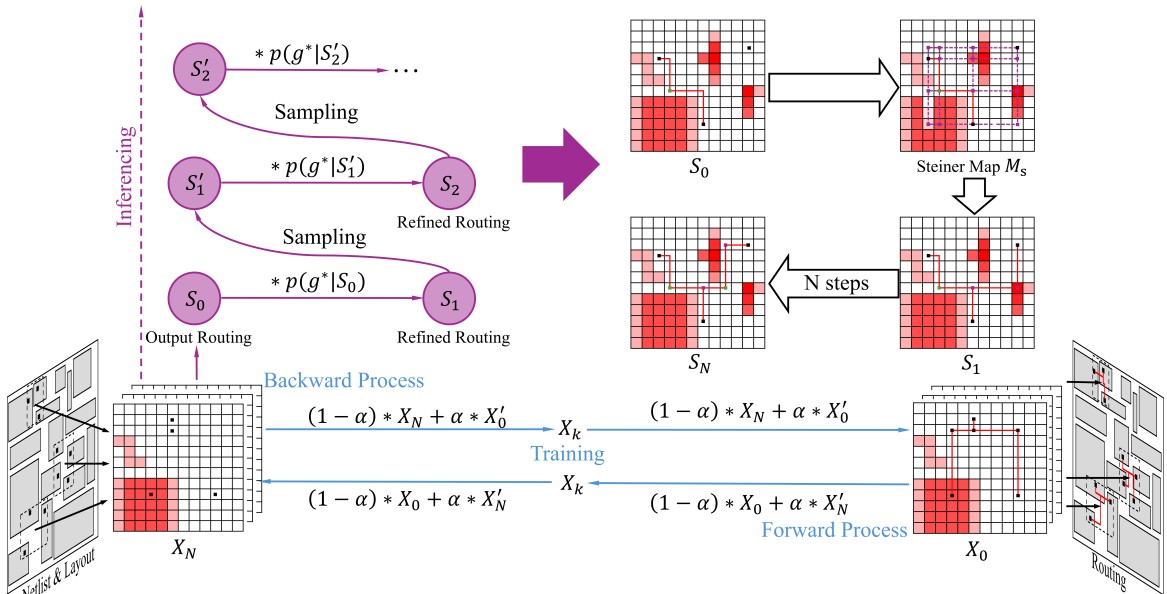

*Figure 2.* **Overview of DSBRouter.** During training, DSBRouter straightly learns the transition between initial pins with congestion heatmap $\mathbf{X}_N$ and connected routes $\mathbf{X}_0$, in bi-direction. The bottom blue part shows how DSBRouter generates a transition trajectory between two marginal distributions, through a noiser scheduler A.3.2. During the inference stage, DSBRouter performs transitions from initial pins to connected routes guided by the objective as shown in the left purple part. The right part shows that, in every step, DSBRouter constructs the Steiner map, based on the existing predicted routing results $S_0$ to output the suboptimal expected routing results $S_1$. Then the optimal objective $\mathbf{g}^*$ is computed, and the gradient of Equation 11 is employed to refine the generated route with low connectivity. After several rounds of optimization, the final routing results $S_N$ is derived. $\alpha$: coefficient parameter, $X_0'$: predicted $X_0$.

optimizing overflow and wirelength. In general combinatorial optimization problems, it is often very difficult—or even practically infeasible—to directly specify the objective score of a given instance. However, in the Global Routing problem, DSBRouter can output a solution at each inference step to make the computation of the objective score for the given instance possible, even if the routing solution is disconnected. We respectively introduce the details of DSB training and gradient feedback in Sec. 3.2 and Sec. 3.3. Training and inference phases are detailed in Appendix A.6.

### 3.2. Global Routing Learning via Schrödinger Bridge

The SB problem is a famous entropy regularized (OT) problem first introduced in (Schrödinger, 1932) and further discussed in De Bortoli et al. (2021); Tang et al. (2024). Given a reference diffusion $p_{ref} \in \phi_{N+1}$ with finite $N$ steps, the data distribution (i.e. the distribution of routes $p_r$ in this paper) and the prior (i.e. the distribution of initial pins with congestion $p_s$) are often set as known to solve the SB to find the closest diffusion $\pi$ to the reference (w.r.t. Kullback–Leibler divergence on path spaces) which satisfies

$$\pi^* = \arg\min_\pi \{KL(\pi|p_{ref}) : \pi \in \phi_{N+1}, \pi_0 = p_r, \pi_N = p_s\},$$
$$(2)$$

where $\pi_0$ and $\pi_N$ are the marginals of $\pi$ at time step 0 and $N$, respectively. Upon acquiring the optimal solution of $\pi^*$, we can sample $\mathbf{x}_0 \sim p_r$ (e.g., the backward transition) by

initially drawing $\mathbf{x}_N \sim p_s$ and iterate the ancestral sampling $\pi_{t|t+1}(x_t|x_{t+1})$. Conversely, the sampling of $\mathbf{x}_N \sim p_s$ (e.g., the forward transition) is also feasible. To solve the SB problem, many researchers employ the Iterative Proportional Fitting (IPF) (Ruschendorf, 1995) to address it and (Tang et al., 2024) dissect the optimization of the joint density into a series of conditional density optimization problems by introducing the Diffusion technologies:

$$\pi^{2n+1} = \arg\min_\pi \{KL(\pi_{k|k+1}|\pi^{2n}_{k|k+1}) : \pi \in \phi_{N+1}, \pi_N = p_s\},$$
$$\pi^{2n+2} = \arg\min_\pi \{KL(\pi_{k+1|k}|\pi^{2n+1}_{k+1|k}) : \pi \in \phi_{N+1}, \pi_0 = p_r\}.$$
$$(3)$$

To optimize Eq. 3, DSB follows the common practice employed in SGM (Ho et al., 2020; Song & Ermon, 2019) to assume $\pi_{t+1|t}, \pi_{t|t+1}$ as Gaussian distributions and model the bi-directional transitions. The training loss of DSB is:

$$\mathcal{L}_{B^n_{k+1}} = \mathbb{E}_{(\mathbf{x}_k, \mathbf{x}_{k+1}) \sim p^n_{k,k+1}} \left[ \|B^n_{k+1}(\mathbf{x}_{k+1}) - \mathbf{x}_{k+1} + \nu_1\|^2 \right],$$
$$\mathcal{L}_{\tilde{F}^{n+1}_k} = \mathbb{E}_{(\mathbf{x}_k, \mathbf{x}_{k+1}) \sim q^n_{k,k+1}} \left[ \|F^{n+1}_k(\mathbf{x}_k) - \mathbf{x}_k + \nu_2\|^2 \right],$$
$$(4)$$

where $\nu_1 = F^n_k(\mathbf{x}_{k+1}) - F^n_k(\mathbf{x}_k)$, $\nu_2 = B^n_{k+1}(\mathbf{x}_k) - B^n_{k+1}(\mathbf{x}_{k+1})$, $q^n = \pi^{2n}$ and $p^n = \pi^{2n+1}$ denote the forward and backward joint densities, respectively. More specifically, $q^n_{k+1|k}(\mathbf{x}_{k+1}|\mathbf{x}_k) = \mathcal{N}(\mathbf{x}_{k+1}; \mathbf{x}_k + \gamma_{k+1}f^n_k(\mathbf{x}_k), 2\gamma_{k+1}\mathbf{I})$ is the forward process and $p^n_{k|k+1}(\mathbf{x}_k|\mathbf{x}_{k+1}) = \mathcal{N}(\mathbf{x}_k; \mathbf{x}_{k+1} + \gamma_{k+1}b^n_{k+1}(\mathbf{x}_{k+1}), 2\gamma_{k+1}\mathbf{I})$ is the backward

**Algorithm 1** Expectation Route Generate

1: **Input:** Initial pins $\eta_0$; route distribution $\mathbf{x}_{k+1}$ at the timestamp $k$;
2: **Output:** Expected route distribution $\mathbf{S}_1$;
3: Initialize $\mathbf{S}^j = \emptyset$;
4: Compute predicted route distribution $\mathbf{x}_k = p_\theta(\mathbf{x}_k|\mathbf{x}_{k+1})$;
5: **for** instance $j \in x_k$ **do**
6:     Extract routing result $\mathbf{S}^j$ for instance $j$;
7:     Extract hubs $h^j$ (Du et al., 2023) in $\mathbf{S}^j$ and merge $h^j$ with $\eta_0^j$ to $\eta_0^j \longleftarrow \eta_0^j \cup h^j$;
8:     **for** combation $c(i,e)$ in $\eta^j(\mathbf{x}_k)$ where $i \in \eta^j(\mathbf{x}_k) \wedge e \in \eta^j(\mathbf{x}_k) \wedge i \neq e$ **do**
9:       **if** $i.x \neq e.x$ or $i.y \neq e.y$ **then**
10:         $\eta^j(\mathbf{x}_k) \longleftarrow \eta^j(\mathbf{x}_k) \cup (i.x, e.y) \cup (e.x, i.y)$;
11:       **end if**
12:     **end for**
13:     Construct the steiner map $\mathbf{M}_s^j$ utilizing $\eta^j(\mathbf{x}_k)$;
14:     Obtain the RST $\mathbf{S}_1^j$ from $\mathbf{M}_s^j$ through algorithm 3;
15:     Update $\mathbf{S}_1 \longleftarrow \mathbf{S}_1 \cup \mathbf{S}_1^j$;
16: **end for**
17: **Return** $\mathbf{S}_1$.

---

**Algorithm 2** Evaluation-Based Guidance

1: **Input:** Expected route distribution $\mathbf{S}_1$; Predicted route distribution $\mathbf{x}_k$ at timestamp $k$; Noiser $N$;
2: **Output:** Noiser distribution $\mathbf{x}_{k+1}$;
3: Obtain $E_{\mathbf{x}_k \sim p_r(\mathbf{x}_k|\mathbf{x}_{k+1})}^o(\eta(\mathbf{x}_k))$ from $\mathbf{S}_1$ and $O(\eta(\mathbf{x}_k))$ from $\mathbf{x}_k$;
4: Compute $p(\mathbf{g}^*|x_k)$ as $\exp([\nabla_{\mathbf{x}_{k+1}}(E_{\mathbf{x}_{k+1} \sim p_r(\mathbf{x}_{k+1}|\mathbf{x}_{k+2})}^o(\eta(\mathbf{x}_{k+1})) - O(\eta(\mathbf{x}_{k+1})))]^\intercal \mathbf{x}_k)$;
5: Update $\mathbf{x}_k$ with Proposition 3.1;
6: $\mathbf{x}_{k+1} = N(\mathbf{x}_k)$.
7: **Return** $\mathbf{x}_{k+1}$.

---

process, where $f_k^n(x)$ and $b_{k+1}^n(\mathbf{x}_{k+1})$ are drift functions. Normally, DSB uses two separate neural networks to approximate $B_{\theta^1}(k, \mathbf{x}) \approx B_k^n(\mathbf{x}) = \mathbf{x} + \gamma_k b_k^n(\mathbf{x})$ and $F_{\theta^2}(k, \mathbf{x}) \approx F_k^n(\mathbf{x}) = \mathbf{x} + \gamma_{k+1} F_k^n(\mathbf{x})$, $\theta^1$ and $\theta^2$ denote the network parameters. In DSBRouter, we employ the terminal reparameterized DSB used in Tang et al. (2024) to tailor loss by Eq. 4 into:

$$\mathcal{L}_{\tilde{B}_{k+1}^n} = \mathbb{E}_{(\mathbf{x}_0,\mathbf{x}_{k+1}) \sim p_{0,k+1}^n}\left[\|\tilde{B}_{k+1}^n(\mathbf{x}_{k+1}) - \mathbf{x}_0\|^2\right],$$

$$\mathcal{L}_{\tilde{F}_k^{n+1}} = \mathbb{E}_{(\mathbf{x}_k,\mathbf{x}_N) \sim q_{k,N}^n}\left[\|\tilde{F}_k^{n+1}(\mathbf{x}_k) - \mathbf{x}_N\|^2\right]. \quad (5)$$

In practice, DSBRouter uses two uvit-b based SGM to approximate the transitions between $p_s$ and $p_r$. For $(2n+1)$-th epoch of DSB, we refer to the optimization of the backward network $B$ and we optimize the forward network in $F$ with $(2n+2)$th epoch. During the inference stage, we focus on

the backward process (i.e., $p_s \to p_r$), and use the sampling method in Tang et al. (2024):

$$p_{\theta^1}^n(\mathbf{x}_k|\mathbf{x}_{k+1}) = \mathcal{N}(\mathbf{x}_k; \mu_{k+1}^n(\mathbf{x}_{k+1}, \mathbf{x}_0), \sigma_{k+1}\mathbf{I}),$$

$$\mu_{k+1}^n(\mathbf{x}_{k+1}, \mathbf{x}_0) \approx \mathbf{x}_{k+1} + \frac{\gamma_{k+1}}{\bar{\gamma}_{k+1}}(\mathbf{x}_0 - \mathbf{x}_{k+1}), \quad (6)$$

$$\sigma_{k+1} = \frac{2\gamma_{k+1}\bar{\gamma}_k}{\bar{\gamma}_{k+1}}.$$

### 3.3. DSBRouter with Gradient Feedback

The incorporation of objective optimization is necessary and important, which enables the direct involvement of objective and effective search over the solution space towards minimizing the score. Note that our task focus on the backward process $p_{\theta^1}^n(\mathbf{x}_k|\mathbf{x}_{k+1}) : p_s \to p_r$. At each step $k$, the model is to estimate $p_\theta(\mathbf{x}_k|\mathbf{x}_{k+1})$. For the purpose of objective optimization, we aim to estimate $p_\theta(\mathbf{x}_k|\mathbf{x}_{k+1}, \mathbf{g}^*)$ where $\mathbf{g}^*$ is the optimal objective score given route distribution $\bar{\mathbf{x}}_k$:

$$\mathbf{g}^* = \arg\min_{\bar{\mathbf{x}}_k} \mathbb{S}(\bar{\mathbf{x}}_k) \quad (7)$$

to effectively guide the backward inference process toward $\mathbf{p}_r^* = \arg\min_{\bar{\mathbf{x}}_k} \mathbb{S}(\bar{\mathbf{x}}_k)$. We will later show that this guidance can be realized by the model trained in Sec. 3.2 without relying on additionally trained nets.

**Objective.** As the generated samples $\bar{\mathbf{x}}_k \sim p_r(\mathbf{x}_k|\mathbf{x}_{k+1})$ at any given step $k$ are not guaranteed to satisfy the constraints, we propose an easy RSMT algorithm 3 to correct the predicted routes so that we can obtain the expectation of overflow $E_{\bar{\mathbf{x}}_k \sim p_r(\bar{\mathbf{x}}_k|\mathbf{x}_{k+1})}^o(\eta(\bar{\mathbf{x}}_k))$ and wirelength $E_{\bar{\mathbf{x}}_k \sim p_r(\bar{\mathbf{x}}_k|\mathbf{x}_{k+1})}^w(\eta(\bar{\mathbf{x}}_k))$ given $\bar{\mathbf{x}}_k \sim p_r(\bar{\mathbf{x}}_k|\mathbf{x}_{k+1})$ where $\eta(\bar{\mathbf{x}}_k)$ represents pins in given routes $\bar{\mathbf{x}}_k$. The implementation of the algorithm is detailed in Appendix A.6. Then the objective function can be defined as:

$$\mathbb{S}(\bar{\mathbf{x}}_k) = |E_{\bar{\mathbf{x}}_k \sim p_r(\bar{\mathbf{x}}_k|\mathbf{x}_{k+1})}^o(\eta(\bar{\mathbf{x}}_k)) - O(\eta(\bar{\mathbf{x}}_k)) + c(\bar{\mathbf{x}}_k)|, \quad (8)$$

where $c(\bar{\mathbf{x}}_\mathbf{k}) = E_{\bar{\mathbf{x}}_k \sim p_r(\bar{\mathbf{x}}_k|\mathbf{x}_{k+1})}^w(\eta(\bar{\mathbf{x}}_k)) - W(\eta(\bar{\mathbf{x}}_k))$ represents the penalty function that focuses on minimizing the wirelength. $O(\eta(\bar{\mathbf{x}}_k))$ and $W(\eta(\bar{\mathbf{x}}_k))$ represent the overflow and wirelength of predicted samples $\bar{\mathbf{x}}_k$ output by the model at timestamp $k + 1$. To enhance the model's ability to generate highly connected probability matrices with low overflow, at step $k$, we aim to bridge the gap between the overflow $O(\eta(\bar{\mathbf{x}}_k))$ and the expected overflow $E_{\bar{\mathbf{x}}_k \sim p_r(\bar{\mathbf{x}}_k|\mathbf{x}_{k+1})}^o(\eta(\bar{\mathbf{x}}_k))$ given density $\bar{\mathbf{x}}_k \sim p_r(\bar{\mathbf{x}}_k|\mathbf{x}_{k+1})$ with the gap between $W(\eta(\bar{\mathbf{x}}_k))$ and $E_{\bar{\mathbf{x}}_k \sim p_r(\bar{\mathbf{x}}_k|\mathbf{x}_{k+1})}^w(\eta(\bar{\mathbf{x}}_k))$ as the penalty.

**Evaluation-based Guidance in Backward Transitions.** With the optimization objective prepared, we aim to estimate $p_\theta(\mathbf{x}_k|\mathbf{x}_{k+1}, \mathbf{g}^*)$. While $p_\theta(\mathbf{x}_k|\mathbf{x}_{k+1}, \mathbf{g}^*)$ can be estimated

*Table 2.* **Correctness rate and generation time on ISPD07 benchmark.** Comparison with 3 hubrouter variants w/o RL as post-process.

| METRIC | CASE | HUBROUTER (VAE) | HUBROUTER (GAN) | HUBROUTER (DPM) | DSBROUTER |
|---|---|---|---|---|---|
| **Correctness Rate** | SMALL-4 | **0 ± 0** | **0.44 ± 0.009** | **0.38 ± 0.008** | **1.000** |
| | SMALL | **0 ± 0** | **0.09 ± 0.003** | **0.06 ± 0.005** | **1.000** |
| | LARGE-4 | **0 ± 0** | **0 ± 0** | **0 ± 0** | **1.000** |
| | LARGE | **0 ± 0** | **0 ± 0** | **0 ± 0** | **1.000** |
| **Wirelength Ratio** | SMALL-4 | 1.101 ± 0.018 | **1.012 ± 0.003** | 1.060 ± 0.010 | 1.015 |
| | SMALL | 1.041 ± 0.004 | 1.002 ± 0.001 | 1.172 ± 0.011 | **1.001** |
| | LARGE-4 | 1.114 ± 0.036 | 1.005 ± 0.002 | 1.100 ± 0.019 | **1.001** |
| | LARGE | 1.044 ± 0.011 | **1.001 ± 0.000** | 1.244 ± 0.022 | 1.450 |
| **Generation Time (GPU Sec)** | SMALL-4 | **6.11 ± 0.23** | 6.96 ± 0.07 | 742.33 ± 4.41 | 2643 |
| | SMALL | **8.88 ± 0.13** | 9.10 ± 0.15 | 739.04 ± 8.20 | 2671 |
| | LARGE-4 | **7.39 ± 0.41** | 8.01 ± 0.09 | 741.09 ± 3.19 | 2687 |
| | LARGE | **11.01 ± 0.11** | 12.34 ± 0.26 | 742.11 ± 4.20 | 2571 |

via the following proposition 3.1, which is adapted in the classifier guidance technique (Dhariwal & Nichol, 2021):

**Proposition 3.1.** *The optimization-enforced denoising probability estimation $p_\theta(\mathbf{x}_k|\mathbf{x}_{k+1}, \mathbf{g}^*)$ equals to $Z p_\theta(\mathbf{x}_k|\mathbf{x}_{k+1})p(\mathbf{g}^*|\mathbf{x}_k)$ ($Z$ is a normalizing constant).*

Though DSB is not a denoising-based model, this proposition still works well as Ho & Salimans (2021) points out that guidance can also be applied to the SDE process and DSB uses SGM (Song et al., 2021b) as the backbone. The proof is listed in Appendix A.5. While $p_\theta(\mathbf{x}_k|\mathbf{x}_{k+1})$ can directly obtained from the trained neural network, the main challenge is to estimate $p(\mathbf{g}^*|\mathbf{x}_k)$. Since $\mathbf{x}_k$ is not accessible at step $k+1$, we apply Taylor expansion to approximate $p(\mathbf{g}^*|\mathbf{x}_{k+1})$ around $\mathbf{x}_k = \mathbf{x}_{k+1}$, given that $\mathbf{x}_k \sim \mathbf{x}_{k+1}$:

$$\log p(\mathbf{g}^*|\mathbf{x}_k) \approx$$
$$\log p(\mathbf{g}^*|\mathbf{x}_{k+1}) + [\nabla_{\mathbf{x}_{k+1}} \log p(\mathbf{g}^*|\mathbf{x}_{k+1})]^\mathsf{T}(\mathbf{x}_k - \mathbf{x}_{k+1})$$
$$= [\nabla_{\mathbf{x}_{k+1}} \log p(\mathbf{g}^*|\mathbf{x}_{k+1})]^\mathsf{T}\mathbf{x}_k + \log p(\mathbf{g}^*|\mathbf{x}_{k+1})$$
$$- [\nabla_{\mathbf{x}_{k+1}} \log p(\mathbf{g}^*|\mathbf{x}_{k+1})]^\mathsf{T}\mathbf{x}_{k+1}, \quad (9)$$

where $\log p(\mathbf{g}^*|\mathbf{x}_{k+1}) - [\nabla_{\mathbf{x}_{k+1}} \log p(\mathbf{g}^*|\mathbf{x}_{k+1})]^\mathsf{T}\mathbf{x}_{k+1}$ is irrelevant to $\mathbf{x}_k$. By exponentiation, we have:

$$p(\mathbf{g}^*|\mathbf{x}_k) \propto \exp([\nabla_{\mathbf{x}_{k+1}} \log p(\mathbf{g}^*|\mathbf{x}_{k+1})]^\mathsf{T}\mathbf{x}_k). \quad (10)$$

To determine $p(\mathbf{g}^*|\mathbf{x}_{k+1})$, we use energy-based model (LeCun et al., 2006) to describe the density $p(\mathbf{g}^*|\mathbf{x}_{k+1})$ and use former part of the objective of Eq. 8 as the energy function:

$$\mathbb{E}(E, \mathbf{x}_k) = E^o_{\mathbf{x}_k \sim p_r(\mathbf{x}_k|\mathbf{x}_{k+1})}(\eta(\mathbf{x}_k)) - O(\eta(\mathbf{x}_k)). \quad (11)$$

It measures how closely $O(\eta(\mathbf{x}_k))$ aligns with expectation $E^o_{\mathbf{x}_k \sim p_r(\mathbf{x}_k|\mathbf{x}_{k+1})}(\eta(\mathbf{x}_k))$. This design aligns with our goal of optimizing the overflow naturally Then, Gibbs distribution is employed to tailor the probability density:

$$p(\mathbf{g}^*|\mathbf{x}_{k+1}) = \frac{\exp(-\mathbb{E}(E, \mathbf{x}_{k+1}))}{\int_{E'} \exp(-\mathbb{E}(E', \mathbf{x}_{k+1}))}. \quad (12)$$

Set $Z = \int_{E'} \exp(-\mathbb{E}(E', x_{k+1}))$, then we have

$$\log p(\mathbf{g}^*|\mathbf{x}_{k+1}) = \quad (13)$$
$$E^o_{\mathbf{x}_{k+1} \sim p_r(\mathbf{x}_{k+1}|\mathbf{x}_{k+2})}(\eta(\mathbf{x}_{k+1})) - O(\eta(\mathbf{x}_{k+1})) - \log Z.$$

The gradient of Eq. 12 can be obtained as

$$\nabla_{\mathbf{x}_{k+1}} \log p(\mathbf{g}^*|\mathbf{x}_{k+1}) = \quad (14)$$
$$\nabla_{\mathbf{x}_{k+1}} (E^o_{\mathbf{x}_{k+1} \sim p_r(\mathbf{x}_{k+1}|\mathbf{x}_{k+2})}(\eta(\mathbf{x}_{k+1})) - O(\eta(\mathbf{x}_{k+1}))).$$

Thus, in this framework, $p(\mathbf{g}^*|\mathbf{x}_k)$ can be estimated as $p(\mathbf{g}^*|\mathbf{x}_k) \propto \exp([\nabla_{\mathbf{x}_{k+1}}(E^o_{\mathbf{x}_{k+1} \sim p_r(\mathbf{x}_{k+1}|\mathbf{x}_{k+2})}(\eta(\mathbf{x}_{k+1})) - O(\eta(\mathbf{x}_{k+1})))]^\mathsf{T}x_k)$ by Eq. 10 and 13. Finally, the guided backward transition is achieved by Proposition 3.1.

## 4. Experiment and Analysis

### 4.1. Datasets and Setups

**Datasets.** For training, We use ISPD07 benchmarks (Nam et al., 2007) to build the marginal distribution $p_s$ and nthurouter (Chang et al., 2008) to perform routing to construct distribution $p_r$. The size of figures in the training set is fixed to $64 \times 64$, following the settings in Du et al. (2023). Additionally, we also introduce the ISPD98 routing benchmarks (Alpert, 1998) to perform global routing and compare metrics between different methods.

**Metrics.** For ISPD07 routing benchmarks, we select the metrics: correctness rate (Crrt, the percentage of routes where all pins are connected within a single route), wirelength ratio (WLR (Cheng et al., 2022), the ratio of the generated route length to the GT length), generation time (taken to generate the route), and connectivity ratio (percentage of the routes of complete connection). In the experiments involving ISPD98, we use wirelength (WL), overflow (OF), and runtime as metrics.

**Baselines.** Three traditional routers (Geosteiner (Juhl et al., 2018), Labyrinth (Kastner et al., 2002), Flute (Wong &

*Table 3.* **Wirelength (WL) & overflow (OF) on ISPD98 & ISPD07.** Comparison of 3 classical global routing and 4 ml-based methods.

| METRICS | MODEL | IBM01 | IBM02 | IBM03 | IBM04 | IBM05 | ADA03 | ADA04 |
|---|---|---|---|---|---|---|---|---|
| WL | GEOSTEINER[*] | **60142** | **165863** | **145678** | **162734** | **409709** | 9330748 | 8865643 |
| | LABYRINTH[*] | 75909 | 201286 | 187345 | 195856 | 420581 | - | - |
| | FLUTE+ES[*] | 61560 | 168841 | 146819 | 167233 | 412816 | 9422071 | 8883791 |
| | HR-VAE | $64703 \pm 1498$ | $176492 \pm 6830$ | $159968 \pm 3281$ | $179895 \pm 5274$ | $434942 \pm 2916$ | - | - |
| | HR-DPM | $66464 \pm 1586$ | $190588 \pm 2337$ | $168454 \pm 2486$ | $183696 \pm 1736$ | $475820 \pm 5516$ | - | - |
| | HR-GAN | $61056 \pm 151$ | $167545 \pm 236$ | $147050 \pm 208$ | $164298 \pm 326$ | $411857 \pm 472$ | 9347032 | 8879952 |
| | NEURALSTEINER[*] | 61735 | 170405 | 148036 | 166648 | 415684 | 9459117 | 9003952 |
| | DSBROUTER(OURS) | 61435 | 174016 | 152862 | 163942 | 420464 | 29478326 | 24276147 |
| OF | GEOSTEINER[*] | 3342 | 7399 | 3944 | 7420 | 401 | 142254 | 45050 |
| | FLUTE+ES[*] | 3100 | 7121 | 3699 | 6889 | 317 | 136661 | 41996 |
| | HR-VAE | $4721 \pm 667$ | $9919 \pm 801$ | $7311 \pm 692$ | $10433 \pm 1299$ | $909 \pm 106$ | - | - |
| | HR-DPM | $4933 \pm 700$ | $14117 \pm 1309$ | $9344 \pm 818$ | $11471 \pm 871$ | $2390 \pm 126$ | - | - |
| | HR-GAN | $3491 \pm 64$ | $7481 \pm 31$ | $4010 \pm 42$ | $7551 \pm 22$ | $419 \pm 7$ | 142119 | 45411 |
| | NEURALSTEINER[*] | 2200 | 3800 | 2100 | 2700 | 18 | 728 | 97 |
| | DSBROUTER(OURS) | **1430** | **0** | **4** | **10** | **0** | **0** | **0** |

[*] EXPERIMENTAL RESULTS CITED FROM (LIU ET AL., 2024)

*Table 4.* Training epochs of DSBRouter and inference time (seconds) with varying inferencing steps on ibm01.

| Steps | Training (epochs) | Inference (s) | OF | WL |
|---|---|---|---|---|
| 10 | 64 | 2774 | 924 | 64237 |
| 24 | 64 | 4378 | 784 | 68982 |
| 50 | 64 | 10881 | **421** | 70172 |
| 24 | 192 | 4491 | 1428 | **61433** |

*Table 5.* **Module ablation in DSBRouter.** Comparison of the unchanged DSBRouter with modified models removing gradient guidance (GD) module and neural network (NN), respectively.

| METRICS | MODEL | IBM01 |
|---|---|---|
| OF | WITHOUT GD | 3561 |
| | WITHOUT NN | 5971 |
| | DSBROUTER | **1430** |
| WL | WITHOUT GD | 61003 |
| | WITHOUT NN | 75975 |
| | DSBROUTER | **61435** |

Chu, 2008) and ES (Chu & Wong, 2005)) as well as two ML-based state-of-the-art methods (HubRouter (Du et al., 2023) and NeuralSteiner (Liu et al., 2024)) are employed as baselines. However, Neural's preprocessing steps using CUGR (Liu et al., 2020) and the corresponding core code have not been made publicly available. Consequently, for NeuralSteiner, we only report the results provided in its original paper (Liu et al., 2024).

**Other Implementation Details.** Details of training/test datasets and other protocols are given in Appendix A. Additional results are listed in Appendix B.

### 4.2. Correctness/Connectivity on Unconnected Cases

We conducted the Crrt and WLR evaluation on the same test cases from part of ISPD07 benchmarks and compared DSBRouter with three different structures of HubRouter

without RL as post-processing. For HubRouter, we repeat the experiments 3 times and present the mean values. The ISPD07 benchmarks, outside the training set, are divided into 'small-4', 'small', 'large-4', and 'large'. '4' in their names represents no more than 4 pins in the chip. 'small' and 'large' represent the Half-perimeter wirelength of the net is less or more than 16. Additionally, we cancel RL as post-processing while using the generated routes as the terminal distribution for HubRouter. As an e2e model, DS-BRouter not only maintains strong connectivity but also generates competitively shorter routes. Tab. 2 highlights DSBRouter's ability to generate routes with high Crrt, all 100% connected across all tested cases. However, all variants of HubRouter achieve an average connectivity ratio of only 27% in the smallest-scale route generation (small-4). In other route generation cases, the connectivity ratio is approximately or exactly 0. And the third line in Tab. 2 shows DSBRouter's ability to generate routes in low WLR. Due to the gradient recomputation and multiple inferencing steps, DSBRouter is behind in generation time. It needs to be mentioned that we do not include PRNet (Cheng et al., 2022) as it shows poor connectivity in (Du et al., 2023).

### 4.3. Routing Results on Real-World Benchmarks

WL, OF, and generation time are evaluated on ISPD98 (ibm01-05) and ISPD07 (adaptec 3 and 4) across various methods. As shown in Tab. 3, DSBRouter significantly reduces the total overflow compared to all other tested methods. Compared with the SOTA ML-based method NeuralSteiner, DSBRouter hits an average reduction of $90.4\%$ and up to $100\%$ on ibm02, ibm05, ada03, and ada04. For wirelength, DSBRouter has better performance on median and small-scale benchmarks. DSBRouter does not incur too many additional nets on ISPD98, maintaining it within $3.2\%$ compared with NeuralSteiner and even outperforming NeuralSteiner on ibm01 and ibm04. However, on ISPD07, DSBRouter produces a longer wirelength compared to Neu-

*Table 6.* **Generation time on ISPD98 and ISPD07 benchmark.** Generation time is evaluated across all tested methods.

| METRIC | MODEL | IBM01 | IBM02 | IBM03 | IBM04 | IBM05 | ADA03 | ADA04 |
|---|---|---|---|---|---|---|---|---|
| | GEOSTEINER | **1.00** | **2.25** | **1.66** | **2.10** | **3.70** | 331.23 | 266.51 |
| | LABYRINTH | 7.11 | 11.08 | 11.61 | 42.03 | 12.70 | - | - |
| | FLUTE+RES | 3.14 | 4.90 | 5.88 | 15.49 | 7.88 | 403.61 | 371.11 |
| TIME | HR-VAE | $10.14 \pm 0.07$ | $9.81 \pm 0.12$ | $11.20 \pm 0.05$ | $10.98 \pm 0.11$ | $12.39 \pm 0.30$ | - | - |
| | HR-DPM | $1833.48 \pm 42.11$ | $2816.77 \pm 22.21$ | $3009.18 \pm 19.00$ | $3842.61 \pm 24.41$ | $4191.71 \pm 36.29$ | - | - |
| | HR-GAN | $39.22 \pm 1.03$ | $44.41 \pm 0.92$ | $49.79 \pm 1.99$ | $68.11 \pm 3.38$ | $76.28 \pm 3.42$ | 1442.71 | 1501.11 |
| | DSBROUTER(OURS) | 4491 | 5667 | 8418 | 10745 | 11313 | 115438 | 125589 |

ralSteiner. We declare that this is reasonable because the ISPD07 test set (ada03 and ada04) we choose includes a massive number of pins and even contains nets with an HPWL exceeding 2000, and thus, it is much larger in scale than the training set. The evaluation-based guidance used by DSBRouter during sampling imposes a constraint on optimizing overflow, which may allow it to bypass congested areas and generate routes with longer WL but lower OF. In terms of generation time, as shown in Tab. 6, DSBRouter lags behind other methods, which we attribute to the longer sampling steps required by DSB during inference.

### 4.4. Ablation Study

**Influence of EG and NN.** To study the influence of the introduced evaluation-based guidance (EG) module in DS-BRouter, we ablate the guidance module from the model architecture. Additionally, to study the role of predicted routes at any timestamp during inference of DSB, we also ablate the output of the neural network (NN). The modified models are tested in comparison with the unchanged DSBRouter on ibm01. We state that the modified model without EG cannot guarantee connectivity of the generated route in the testing case so we employ Geosteiner as the post-processing to complete the routing. Tab. 5 indicates that OF is affected by both EG and the output of the NN. We can witness a significant increase both in WL and OF from the output of model without NN. We also claim that the model without EG but with Geosteiner as post-processing can generate a shorter route all due to the inherent devotion of Geosteiner optimizing WL as an increase of OF is confirmed. The results on ibm01 imply that both EG and DSB can help DSBRouter acquire congestion-avoiding routes.

**Influence of inferencing steps.** In DSB, the forward and backward processes share the same number of inferencing steps. The number of inferencing steps determines the step size required for the DSB to transform the marginal distribution $p_s$ (or $p_r$) to $p_r$ (or $p_s$). We conduct evaluations on the inferencing steps of DSB to study the effect of inferencing steps on the generation time, OF, and WL. We fixed the training (inferencing) steps to 64 (192 in all other experiments mentioned above) and set the inferencing steps to 10, 24 (the default value in all other experiments mentioned above), and 50, respectively, to train three different DSB

backbones. Results in Tab. 4 indicate that as the number of inferencing steps increases, inference time also increases, from 2774 to 10881 as inferencing steps increase from 10 to 50. As the number of inferencing steps increases, the inferencing time naturally increases as well. When the number of inferencing steps increases from 10 to 50, the inferencing time increases from 2774 to 10881. However, under the same training step size, the effect of increasing the number of inferencing steps is not entirely negative. As shown in the first three rows of Tab. 4, with the same training depth, as the number of inferencing steps increases, DSBRouter can generate routes with lower OF, but correspondingly, the WL also increases. As the training depth increases, DSBRouter generates routes with shorter WL at the cost of OF.

## 5. Further Discussion and Conclusions

**End-to-End design for global routing.** We have investigated the potential of DSB for the global routing task in VLSI. In supervised scenarios, DSB can directly establish a bidirectional transformation bridge between data Distribution and prior Distribution. Meanwhile, we introduce the goal-oriented sampling with evaluation-based guidance technique from SGM into the inference phase of DSB, enabling DSBRouter to produce high-quality, connected routes even in unsupervised settings. To the best of our knowledge, DSBRouter is the first learning–based, end-to-end router.

**Conclusion.** In this paper, we propose an End-to-End ml-based DSBRouter for the global routing problem in VLSI. DSBRouter is the first End-to-End router to perform routing tasks and outperforms other state-of-art two-stage routers in reduction of overflow by an average of 90%.

**Limitations.** Due to the non-monotonically noise injection strategy and meaningful $p_r, p_s$ in the DSB-based sampling process, accelerated sampling techniques such as DDIM in DPM cannot be directly applied. Also, at each step $k$, computing the $\mathbf{g}^*$ and subsequent gradients based on the network's reparameterized outputs requires substantial computational resources. Consequently, despite employing parallel processing techniques, DSBRouter still suffer from long running time. In the future, we will accelerate the inference with the distillation-based models (Song et al., 2023; Shi et al., 2025b) to improve the efficiency.

## Impact Statement

This paper presents work whose goal is to advance the field of Machine Learning and Artificial Intelligence for Electronic Design Automation (AI4EDA). There are many potential societal consequences of our work, none of which we feel must be specifically highlighted here.

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

# A. Implementation Details

## A.1. Connection between SGMs and Simpliflied Diffusion Schrödinger Bridges

We incorporate the simpiflied diffusion schrödinger Bridge (Tang et al., 2024) as the backbone of our generator in DSBRouter. In this section, we will take an insight into the connection between generative models (especially SGMs) and DSB to pave the way for the tailored gradient search brought into DSB.

In earlier studies (Schrödinger, 1932; Léonard, 2013), it has been demonstrated that the optimal solution to the Schrödinger Bridge problem can be characterized by the following stochastic differential equation (SDE):

$$
\begin{aligned}
d\mathbf{X}_t &= (f(\mathbf{X}_t, t) + g^2(t) \bigtriangledown \log \psi(\mathbf{X}_t, t))dt + g(t)d\mathbf{W}_t, X_0 \sim p_{data}, \\
d\mathbf{X}_t &= (f(\mathbf{X}_t, t) - g^2(t) \bigtriangledown \log \hat{\psi}(\mathbf{X}_t, t))dt + g(t)d\bar{\mathbf{W}}_t, X_T \sim p_{prior},
\end{aligned}
\tag{15}
$$

where $\mathbf{W}_t$ is Wiener process and $\bar{\mathbf{W}}_t$ its time reversal. $\psi, \hat{\psi} \in C^{2,1}([0, T], \mathbb{R}^d)$ are time-varying energy potentials that constrained by the interconnected PDEs:

$$
\begin{cases}
\frac{\partial \psi}{\partial t} = -\bigtriangledown_x \psi^\top f - \frac{1}{2}Tr(g^2 \bigtriangledown_x^2 \psi) \\
\frac{\partial \hat{\psi}}{\partial t} = -\bigtriangledown_x \cdot (\hat{\psi} f) + \frac{1}{2}Tr(g^2 \bigtriangledown_x^2 \hat{\psi}),
\end{cases}
\tag{16}
$$
$$
\text{s.t. } \psi(x, 0)\hat{\psi}(x, 0) = p_{data},
$$
$$
\psi(x, T)\hat{\psi}(x, T) = p_{prior}.
$$

More generally, we can achieve the distribution of SB at time t by:

$$
p_t = \psi(x, t)\hat{\psi}(x, t).
\tag{17}
$$

Score-based Generative Models (SGMs) (Ho et al., 2020; Sohl-Dickstein et al., 2015; Song et al., 2021a; Song & Ermon, 2019) connect two densities through a dual process: a forward process that transitions the data distribution, $p_{data}$, toward a prior distribution $p_{prior}$ and a reverse process, typically guided by neural networks, that converts the prior back to the data distribution. These two processes can be modeled as Markov chains. Given an initial data distribution $p_{data}$ and a target prior distribution $p_{prior}$, the forward process $p_{k+1|k}(x_{k+1}|x_k)$ is designed to transition from $p_0 = p_{data}$ step-by-step t.pproximation of $p_{prior}$. This process generates a sequence $x_{0:N}$ from the (N+1) intermediate steps. This trajectory's joint probability distribution is then formally defined as:

$$
p(x_{0:N}) = p_0(x_0) \prod_{k=0}^{N-1} p_{k+1|k}(x_{k+1}|x_k).
\tag{18}
$$

Through the backward process, the joint density can also be reformulated as a time-reversed distribution:

$$
p(x_{0:N}) = p_N(x_N) \prod_{k=0}^{N-1} p_{k|k+1}(x_k|x_{k+1}),
\tag{19}
$$

however, directly computing $p_{k|k+1}(x_k|x_{k+1})$ is typically challenging. SGM utilizes a simplified approach that regard the forward process as a gradual adding of Gaussian noise:

$$
p_{k+1|k}(x_{k+1}|x_k) = \mathcal{N}(x_{k+1}; x_k + \gamma_{k+1}f_k(x_k), 2\gamma_{k+1}\mathbf{I}).
\tag{20}
$$

It follows that for a sufficiently extensive $N$, the distribution $p_N$ will converge to Gaussian distribution, which we set as $p_{prior}$. Moreover, the temporal inversion in Equation 19 can be analytically approximated (Anderson, 1982; Hyvärinen & Dayan, 2005; Vincent, 2011) as 9. Subsequently, SGM employs neural networks $s_\theta(x_{k+1}, k + 1)$ to approximate the score term $\bigtriangledown \log p_{k+1}(x_{k+1})$, thus the reverse process can be effectively modeled. By sampling $x_N \sim p_{prior}$, followed by iteratively applying ancestral sampling vua $x_k \sim p_{k|k+1}(x_k|x_{k+1})$, culminating in the estimation of $x_0 \sim p_{data}$. While the diffusion and denoising processes modeled in SGM (Song & Ermon, 2019) can also be formulated as continuous-time Markov chains. The forward process can be represented as a continuous-time SDE:

$$
d\mathbf{X}_t = f_t(\mathbf{X}_t)dt + \sqrt{2}d\mathbf{B}_t.
\tag{21}
$$

*Table 7.* **Summary of the test dataset.** We respectively show the scale size, verticalhorizontal capacity, number of nets, and average/maximum number of pins for each net.

| CASE | IBM01 | IBM02 | IBM03 | IBM04 | IBM05 | ADA03 | ADA04 |
|---|---|---|---|---|---|---|---|
| SIZE | $64 \times 64$ | $80 \times 64$ | $80 \times 64$ | $96 \times 64$ | $128 \times 64$ | $774 \times 779$ | $774 \times 779$ |
| CAP.(V/H) | 24/28 | 44/68 | 40/60 | 40/46 | 84/126 | 62/62 | 62/62 |
| NETS | 11507 | 18429 | 21621 | 26163 | 27777 | 466295 | 515304 |
| AVG.PINS | 4.31 | 4.88 | 4.10 | 3.86 | 5.25 | 4.02 | 3.71 |
| MAX.PINS | 42 | 134 | 55 | 46 | 17 | 3713 | 3974 |

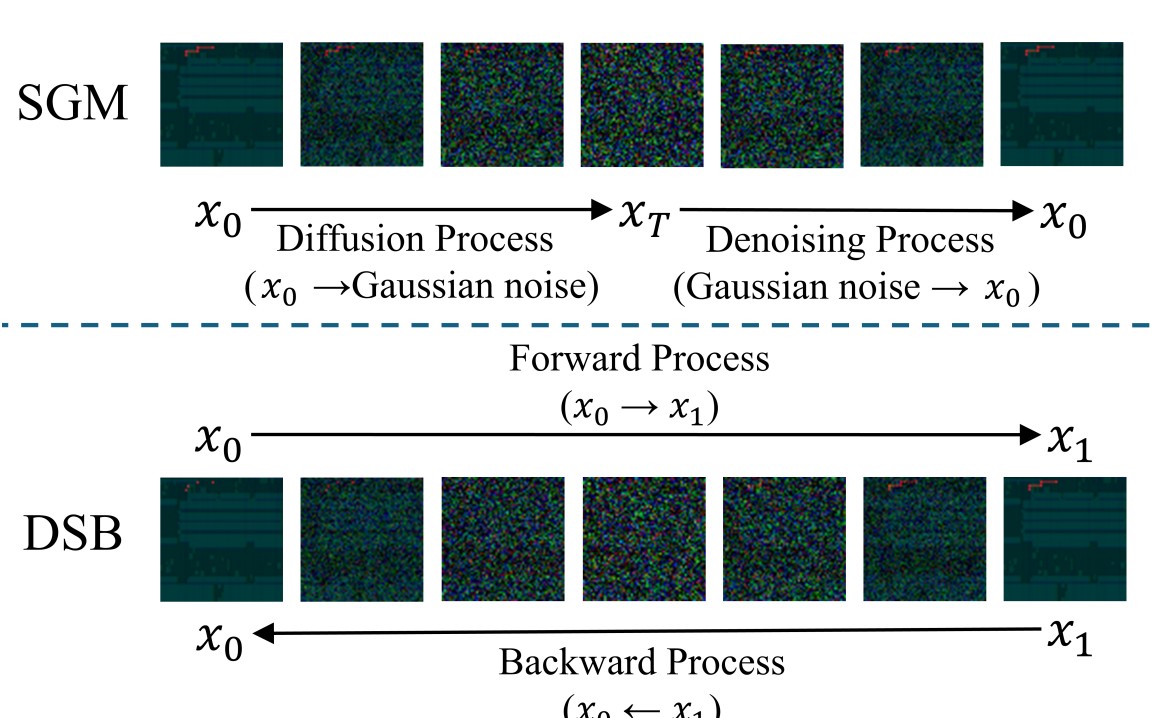

*Figure 3.* **Similarities and differences between SGM and DSB.** In SGMs, the marginal distribution $x_0$ is gradually transformed into an approximate standard Gaussian distribution $x_T$ by introducing niose through SDE. During the inference (denoising) phase, the reverse SDE is then used to recover the original distribution from the approximated Gaussian distribution. In contrast, DSB learns a conditional probability density, thus enabling a direct transition between the two maginal distributions $x_0$ and $x_1$, during inferencing. Regardless of whether SGMs or DSB, the inferencing $x_k$ at timestamp $k$ always depends on the preceding $x_{k-1}$ (or $x_{k+1}$ in DSB) or the original distribution $x_0$ (or $x_N$ in DSB).

Upon close inspection, one can observe that Equation 21 and Equation 15 differ only by the additional non-linear drift term $g^2(t) \triangledown \log \psi(\mathbf{X}_t, t)$. Notably, SGMs may be regarded as a particular instantiation of DSB when the non-linear drift terms are set to zero (i.e. $\psi(\mathbf{X}_t, t) \equiv C$). For Variance Preserving (VP) (Ho et al., 2020) and Variance Exploding (VE) (Song & Ermon, 2019) noise schedule in SGMs, $f(\mathbf{X}_t, t) \equiv -\alpha_t \mathbf{X}_t$ where $\alpha_t \in \mathbb{R}_{\geq 0}$, and the denoising model is essentially sloving Equation 15 using a learnable network.

More general, other dynamic generative models, e.g., Flow Matching (FM) (Lipman et al., 2023), $I^2$SB (Liu et al., 2023), Bridge-TTS (Chen et al., 2023), can be encapsulated within the framework of Equation 15 by selecting appropriate functions for $f(\mathbf{X}_t, t)$ and $\psi(\mathbf{X}_t, t)$. In addition, as depicted in FIG. 3, both DSB and other SGMs, during their inference phase, exhibit actions that can be categorized as predicting the next distribution $x_{k-1}$ (or $x_{k+1}$ in DSB) based on the current distribution $x_k$, which only relies on $x_k$ and is independent of $\mathbf{g}^*$. This means that the derivation (Equation 22) still works in DSB. This unification of dynamic generative models suggests the possibility of a more profound linkage between DSB, SGM and other dynamic generative models, which suggests the rationality to bring the commonly used gradient search scheme in SGM into DSB.

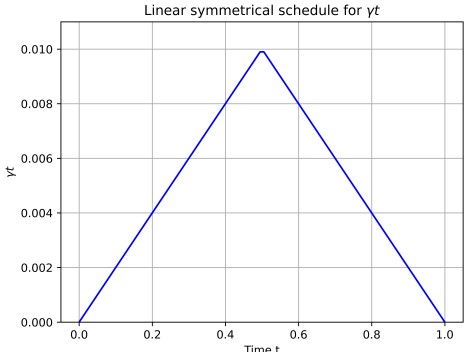

*Figure 4.* Linear schedule for $\gamma_t$

## A.2. Experimental protocols

### A.2.1. DATASETS AND HARDWARE FOR EXPERIMENTS

Real-world Datasets ISPD07 (Nam et al., 2007) and ISPD98 (Alpert, 1998) are employed in this work. In line with (Du et al., 2023), we construct the expert training datasets with low overflow using nthurouter (Chang et al., 2008) to route on parts of ISPD07 benchmarks, including bigblue4, newblue4, newblue5, newblue6 and newblue7. Each case has about 60k samples. Thus the training datasets have a total of nearly 300K samples. We initialize the capacity given by the benchmarks and sequentially route the nets using the results of (Chang et al., 2008). Each time the capacity is updated, a condition image, consisting of the current capacity and the positions of pins to be routed in the next net, is generated. Meanwhile, a ground-truth route image is generated and saved correspondingly. By clipping them to the same scale 64 × 64 (if possible) randomly. For the tested cases in Tab. 2, we choose newblue1, newblue2, bigblue1 and bigblue2 from ISPD07, outside the training sets, with a total 10k samples. Summary of tested cases in ISPD98 are shown in Tab. 7. We follow the same processing steps introduced in (Du et al., 2023) to have the tested ISPD98 cases prepared.

Training of the backbone of DSB is conducted on a machine with an Intel Xeon Platinum 8480+ CPU, 8 NVIDIA H800 GPUs, and 2.0TB RAM. All experiments in this work are conducted on a machine with an Intel Xeon Platinum 8480+ CPU, 8 NVIDIA RTX 4090 GPUs, and 460GB RAM.

## A.3. DSBRouter Network Architecture

### A.3.1. BACKBONE

We use the uvit_b architecture, based on the ViT structure (Dosovitskiy et al., 2021), as the backbone for our model. The input to our model is an image of size batch_size×3×64×64. The first channel represents the pin map, the second channel represents the horizontal congestion heatmap, and the third channel represents the vertical congestion heatmap. The final output is also a three-channel image, where the second and third channels remain unchanged, and the first channel contains the predicted map that reveals the locations of all the pins in the predicted route. After being input into the model, the image is first divided into non-overlapping 4×4 patches, resulting in (64/4)×(64/4) patches in total. Each patch is then mapped to a 512-dimensional embedding vector. These encoded vectors are subsequently fed into a Transformer layer (13 layers) to learn and capture spatial and contextual information between patches. Finally, the vector is resized to match the original input size for image output.

### A.3.2. NOISE SCHEDULER

Noise scheduler determines how the data between the two distributions is transformed, which is equivalent to the noise addition method used in SGM. In the DSB framework, both the prior and posterior distributions represent meaningful data distributions, rather than the standard Gaussian distribution. Therefore, the noise intensity $\gamma_t$ cannot monotonically change in either the forward or reverse process. Consequently, in DSB, we adopt the same approach as in (Tang et al., 2024), where the noise scale $\gamma_t$ undergoes a symmetric linear variation: $\gamma_t$ increases first and then decreases linearly, with the magnitude of increase and decrease being the same, as depicted in Fig. 4

### A.3.3. MOEL PARAMETERS

We use a fixed learning rate of $lr = 0.001$ and a batch size of 256. During each epoch, we repeat the training process for each batch 4 times.

## A.4. Baselines

The baselines referred in Tab. 3 are introduced as follows:

1) *GeoSteiner* (Juhl et al., 2018): the optimal RSMT construction solver.

2) *Labyrinth* (Kastner et al., 2002): A classical routing algorithm that explores how the concept of pattern routing can be utilized to guide the router toward a solution that minimizes interconnect delay while preserving the routability of the circuit.

3) *FLUTE* (Wong & Chu, 2008): A fast and accurate RSMT construction method using a look-up table. It is important to note that this approach can achieve the optimal solution for nets with up to 9 degrees.

4) *Eege Shifting* (Chu & Wong, 2005): A fast, practical RSMT-based algorithm that leverages a specialized lookup table for small nets and a refined recursive splitting approach for larger nets.

5) *HubRouter* (Du et al., 2023): A global router for RST construction based on reinforcement learning. The hub is generated using a diffusion model, followed by reinforcement learning for RST construction.

6) *NeuralSteiner* (Liu et al., 2024): A state-of-the-art two-stage global router. The candidate points are predicted using an RCCA-enhanced CNN, and routing is performed using an RST construction algorithm based on a greedy strategy.

## A.5. Proof of Proposition 3.1

**Proof.** The density $p(\mathbf{x}_k|\mathbf{x}_{k+1}, \mathbf{g}^*)$ can be derived by:

$$
\begin{aligned}
p(\mathbf{x}_k|\mathbf{x}_{k+1}, \mathbf{g}^*) &= \frac{p(\mathbf{x}_{k+1}, \mathbf{x}_k, \mathbf{g}^*)}{p(\mathbf{x}_{k+1}, \mathbf{g}^*)} = \frac{p(\mathbf{x}_{k+1}, \mathbf{x}_k, \mathbf{g}^*)}{p(\mathbf{g}^*|\mathbf{x}_{k+1})p(\mathbf{x}_{k+1})} \\
&= \frac{p(\mathbf{g}^*|\mathbf{x}_{k+1}, \mathbf{x}_k)p(\mathbf{x}_k|\mathbf{x}_{k+1})p(\mathbf{x}_{k+1})}{p(\mathbf{g}^*|\mathbf{x}_{k+1})p(\mathbf{x}_{k+1})} \\
&= \frac{p(\mathbf{g}^*|\mathbf{x}_{k+1}, \mathbf{x}_k)p(\mathbf{x}_k|\mathbf{x}_{k+1})}{p(\mathbf{g}^*|\mathbf{x}_{k+1})}
\end{aligned}
\tag{22}
$$

as $p(\mathbf{g}^*|\mathbf{x}_{k+1})$ is not depend on $\mathbf{x}_k$, thus it can be regarded as a constant. Now we can obtain $p_\theta(\mathbf{x}_k|\mathbf{x}_{k+1}, \mathbf{g}^*) \propto Z p_\theta(\mathbf{x}_k|\mathbf{x}_{k+1})p(\mathbf{g}^*|\mathbf{x}_k)$.

## A.6. Algorithms

Algorithm 1 extends the route output by model at timestamp $k$ by expanding routes with poor connectivity to incorporate potential steiner points, thereby generating routes with lower OF. After obtaining the Steiner Map $\mathbf{M}_s$, we have a greedy disjoint-set based RSMT construct method to derive an OF-minimium RST given $\mathbf{M}_s$ and $\mathbf{S}$. The proposed method is detailed in algorithm 3. The training process and inferencing process are detailed as follows:

# B. Additional Results

## B.1. Route Generation Results

Examples of the initial pins, generated routing results by DSB and real routes are depicted in 5.

## B.2. Relative Error on Part of ISPD98 cases

To judge the improvement of gaining on the optimal wirelength rather than the absolute value, we further compare the relative error in Tab. 8, where the relative error is computed as $(WL - LB)/LB$. Here, $LB$ denotes the theoretical lower bound. As shown in the table, the WL promotion of DSBRouter can hit the lowest on the cases of ibm01 and ibm04. However, as the scale increases, the realtive error of DSBRouter also increases dramatically, which leaves improvement space for DSBRouter.

---

**Algorithm 3** RSMT Construct

---

1: **Input:** Steiner Map $\mathbf{M}_s^j$; Routing result $\mathbf{S}^j$;
2: **Output:** OF-minimium RST $\mathbf{S}_1^j$;
3: Initialize disjoint-set $U_1$ given $\mathbf{S}^j$;
4: **if** $len(U_1.find) == 1$ **then**
5:    $\mathbf{S}_1^j \longleftarrow \mathbf{S}^j$;
6: **end if**
7: Initialize disjoint-set $U_2$ by assigning pin $p \in \mathbf{M}_s^j$ to themselves;
8: Construct edges set $(i, e) \in \mathcal{E}; i \in \mathbf{M}_s^j; e \in \mathbf{M}_s^j$ where pin $i$ and pin $e$ are always vertically or horizontally two nearest points;
9: Initialize $\mathbf{S}_1^j$;
10: **while** $len(U_2.find) \neq 1$ **do**
11:    Edge $(i, e) = \underset{OF}{argmin}\mathcal{E}$;
12:    $\mathbf{S}_1^j \longleftarrow \mathbf{S}_1^j \cup (i, e)$;
13:    Update $U_2$;
14: **end while**
15: **Return** $\mathbf{S}_1^j$.

---

**Algorithm 4** Training of DSB

---

1: **Input:** Number of training iterations $iters$; training steps (inference steps) $steps$; repeat times $r_{times}$; batch size $bc$; noise scheduler $N$; Target distribution $p_r$;
2: **Output:** Model Parameters $\theta$;
3: Initialize model parameters $\theta$;
4: **for** $iter = 0$ to $iters$ **do**
5:    Sample a batch $\mathbf{x} \subset p_s$ where $|\mathbf{x}| = bc$;
6:    **for** $r = 0$ to $r_times$ **do**
7:       **for** $s = 0$ to $steps$ **do**
8:          Apply nosie adding to $\mathbf{x}_s = N(\mathbf{x}, s)$;
9:          Obtain target $\mathbf{x}_N$ from $p_r$;
10:          Descend the stochastic gradient loss of 5;
11:       **end for**
12:    **end for**
13: **end for**
14: **Return** Model Parameters $\theta$.

---

*Table 8.* **Relative error on ISPD98.** The routing results of (Juhl et al., 2018) are treated as the theoretical lower bound. Optimal results are in bold.

| MODEL | IBM01 | IBM02 | IBM03 | IBM04 | IBM05 | ADA03 | ADA04 |
|---|---|---|---|---|---|---|---|
| LOWER BOUND | 60142 | 165863 | 145678 | 162734 | 409709 | 9330748 | 8865643 |
| LABYRINTH | 0.262 | 0.213 | 0.286 | 0.203 | 0.026 | - | - |
| FLUTE+RES | 0.023 | 0.017 | **0.007** | 0.027 | 0.007 | 0.009 | 0.002 |
| HR-VAE | $0.075 \pm 0.024$ | $0.064 \pm 0.04$ | $0.098 \pm 0.022$ | $0.105 \pm 0.032$ | $0.061 \pm 0.007$ | - | - |
| HR-DPM | $0.105 \pm 0.026$ | $0.149 \pm 0.014$ | $0.156 \pm 0.017$ | $0.128 \pm 0.010$ | $0.161 \pm 0.013$ | - | - |
| HR-GAN | $0.022 \pm 0.002$ | $\mathbf{0.010 \pm 0.001}$ | $0.009 \pm 0.001$ | $0.009 \pm 0.002$ | $\mathbf{0.005 \pm 0.001}$ | **0.001** | **0.001** |
| NEURALSTEINER | 0.026 | 0.027 | 0.016 | 0.024 | 0.014 | 0.013 | 0.015 |
| DSBROUTER(OURS) | **0.021** | 0.049 | 0.049 | **0.007** | 0.026 | 3.15 | 2.73 |

**Algorithm 5** Sampling Routes with evaluation-based guidance

1: **Input:** Initial distribution $p_s$; inference steps (training steps) $stpes$; Model parameters $\theta$; batch size $bc$; noise scheduler $N$;
2: **Output:** Generated routing distribution $\bar{\mathbf{x}}_N$;
3: Initialize $\bar{\mathbf{x}}_N = \emptyset$;
4: Obtain size $size$ of $p_s$;
5: **for** $size > 0$ **do**
6:     **for** $s = 0$ to $steps$ **do**
7:         **if** $s == 0$ **then**
8:             $\mathbf{x}_{s+1} = \mathbf{x}_s$;
9:             $\mathbf{x}_{s+2} = \mathbf{x}_s$;
10:         **end if**
11:         Obtain routing result $\mathbf{x}_s = p_\theta(\mathbf{x}_s|\mathbf{x}_{s+1})$;
12:         Obtain Expected routes $\mathbf{S}_1$ given $\mathbf{x}_s$ through algorithm 1;
13:         Compute $\nabla_{\mathbf{x}_{s+1}}(E^o_{\mathbf{x}_{s+1}\sim p_r(\mathbf{x}_{s+1}|\mathbf{x}_{s+2})}(\eta(\mathbf{x}_{s+1})) - O(\eta(\mathbf{x}_{s+1})))$;
14:         Obtain $\nabla_{\mathbf{x}_{s+1}} \log p(\mathbf{g}^*|\mathbf{x}_{s+1})$ through equation 14;
15:         Obatin refined routing result $p_\theta(x_s|x_{s+1}, \mathbf{g}^*)$ through proposition 1;
16:         Apply noise adding to $\mathbf{x}_s = N(\mathbf{x}, s)$;
17:         **if** $s \neq steps$ **then**
18:             $\mathbf{x}_{s+1} = \mathbf{x}_s$;
19:         **end if**
20:     **end for**
21:     $\bar{\mathbf{x}}_N \longleftarrow \bar{\mathbf{x}}_N \cup \mathbf{x}_s$;
22: **end for**
23: **Return** $\bar{\mathbf{x}}_N$.

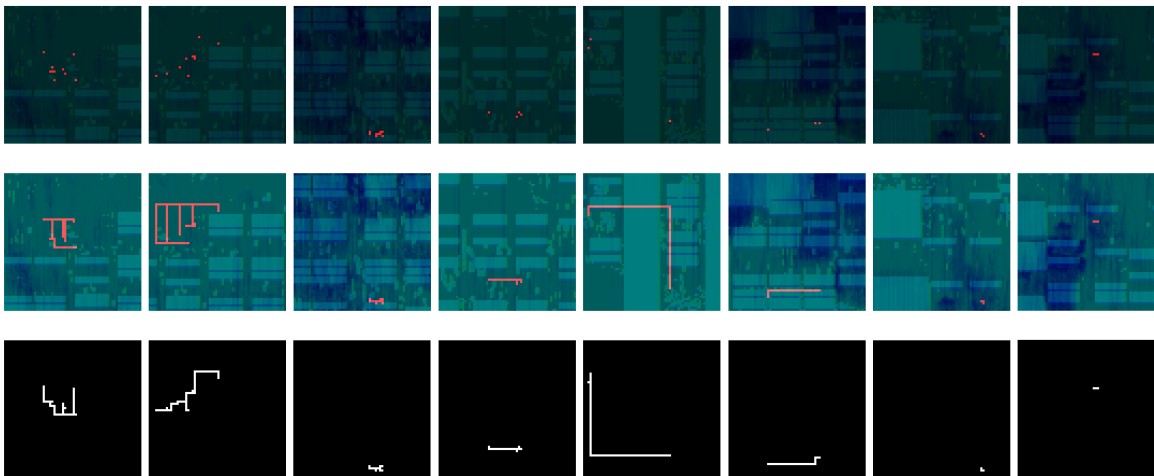

*Figure 5.* Initial pins (first line), Generated routing results by DSB (second line), and the real routes (third line). All were randomly sampled in 'small-4', 'small', 'large-4', and 'large'.

