# OpenReview forum: "DSBRouter: End-to-end Global Routing via Diffusion Schr\"{o}dinger Bridge"
_ICML.cc/2025/Conference — ICML 2025 poster_

### Official Review · Reviewer_4dzK · 2025-03-13

**Overall Recommendation:** 3

**Summary:**

DSBRouter is an end-to-end neural global routing solver based on the Diffusion Schrödinger Bridge (DSB) model, which learns the forward and backward mapping between initial pins and routing results. It achieves state-of-the-art performance in overflow reduction on ISPD98 and parts of ISPD07, with some cases achieving zero overflow without requiring post-processing.

**Claims And Evidence:**

- This is the first work to introduce the DSB technique to the global routing problem to the best of my knowledge.

- This paper introduces instance-wise objective scores, enabling optimization while ensuring general feasibility.

- This paper claims it does not need post-processing. However, from Figure 2, there are refining steps in the inference process.

**Essential References Not Discussed:**

[1] DGR: Differentiable Global Router, DAC 2024.

**Experimental Designs Or Analyses:**

- The evaluation is reasonable. It would be better if the authors could report the mean and standard deviation.

**Methods And Evaluation Criteria:**

- The method is built on the Diffusion Schr ̈odinger Bridge, which can incorporate constraints into the diffusion process.


- The evaluation criteria (Wirelength, overflow, running time) is a general evaluation metric for routing tasks.

**Other Comments Or Suggestions:**

- In table 1, what does the label "△∗" mean?
- According to Fig. 5, the DSB-generated routes should also modified by regulations. which means it is not a pure end-to-end method.
- Typo: Section 4.1, Metircs->Metrics.
- Figure 1, one left-bottom pin is not easy to read.

**Other Strengths And Weaknesses:**

- This work should collect a large batch of routing results as training data, where traditional global routers do not need this process. According to Table 2 and 4, the method is not very efficient.

**Questions For Authors:**

- In Table 3, Why not report the mean and variance?

- Why the size of figures in the training set is fixed to 64 × 64, where the testing set sizes are different.

**Relation To Broader Scientific Literature:**

This work builds on prior global routing methods and diffusion-based models by introducing DSBRouter, which leverages the Diffusion Schrödinger Bridge to ensure connectivity without post-processing. Unlike traditional approaches, it achieves state-of-the-art overflow reduction and incorporates constraints directly into the routing process.

**Theoretical Claims:**

NA

---

> ### Author Rebuttal · Authors · 2025-04-01
>
> > **Weakness 1: This work should collect a large batch of routing results as training data, where traditional global routers do not need this process. According to Table 2 and 4, the method is not very efficient.**
>
> Thanks for your valuable comment. Though traditional global routers indeed do not rely on large datasets for training, we argue that they still exhibit the following issues:
> 1) **Escalating Complexity and Longer Runtime**
> As process nodes continue to shrink and chip sizes grow, traditional methods often demand higher computational resources and longer execution time. Furthermore, these methods rely on continuous updates and refinements from human experts.
> 2) **Heavy Dependence on Manually Crafted Heuristics**
> Traditional routing methods heavily depend on heuristics that are manually specified or tuned, their applicability is limited when the design environment changes.
> 3) **Prone to Local Optima and Complex Multi-Objective Trade-Offs**
> When dealing with highly complex objectives—such as congestion, timing, power, wirelength, IR drop, and other factors—conventional methods struggle to balance all of these goals simultaneously.
>
> By contrast, ML-based methods like DSBRouter yield routing solutions in a feasible runtime, matching or even surpassing traditional routers. They also adapt across diverse routing scenarios and need no expert updates to routing rules.
>
> And we admit that DSBRouter currently faces challenges from relatively longer running time. Several improvements is being considered to address the efficiency of DSBRouter:
> 1) adopt the techniques of the [Consistency Model](https://openreview.net/forum?id=FmqFfMTNnv) to reduce the sampling steps.
> 2) Recomputing the optimization objective every few steps during the inference process, which reduces the computation of evaluation-based guidance.
>
> Although we are still in the process of implementing the aforementioned two improvements, we can estimate that after these enhancements, the runtime of DSBRouter could be reduced by 60%-70%. Once the experiments are finished. The results will be added in our later rebuttal and final version.
>
> > **Suggestion 1: According to Fig. 5, the DSB-generated routes should also modified by regulations. which means it is not a pure end-to-end method.**
>
> We believe the misunderstanding arises from unclear explanation of Figure 5. The first row shows randomly selected initial pins from our supervised dataset. The second row presents the corresponding results generated by DSBRouter, based on the first-row inputs. The third row shows the ground-truth routing results from the dataset, not DSBRouter's outputs.
>
> We also emphasize that DSBRouter is an end-to-end global router. In ML-based methods, if the entire process from input to output is learned and optimized by a single model without losing information, it is considered end-to-end. DSBRouter uses a single model, incorporating evaluation-based guidance in the inference process with no intermediate outputs, making it an end-to-end global router.
>
> > **Suggestion 2, 3 and 4: Typo: Section 4.1, Metircs->Metrics; Figure 1, one left-bottom pin is not easy to read; In table 1, what does the label "△∗" mean?**
>
> $\triangle*$ means PRNet is only scalable for one-shot generation, but not scalable for post-processing. We have add the footnode in our revised paper. The modified sections will be presented in our final version.
>
> > **Question 1: DGR: Differentiable Global Router, should be discussed.**
>
> Thanks for your valuable comment. We noticed this work during our survey, but since their [GitHub repository](https://github.com/search?q=DGR%3A%20Differentiable%20Global%20Router&type=repositories) seems to be private and lacks code to reproduce the results, we did not use it as a baseline. If they release their implementation, we will conduct comparative experiments.
>
> > **Question 2: In Table 3, Why not report the mean and variance?**
>
> For the routing results of GeoSteiner, FLUTE, ES, and NeuralSteiner, we used the experiments from [NeuralSteiner](https://proceedings.neurips.cc/paper_files/paper/2024/hash/e6617714485265b9380a5315bf3ba98f-Abstract-Conference.html). For DSBRouter, we repeated the experiments three times with no observed variance, due to the neural network's determinism and the lack of post-processing in DSBRouter. The variance in HubRouter's results is due to the inherent uncertainty in its RL-based post-processing.
>
> > **Question 3: Why the size of figures in the training set is fixed to 64 × 64, where the testing set sizes are different.**
>
> We believe the misunderstanding is due to the "Size" in Table 7. The "Size" in Table 7 refers to the original grid size of the benchmark, not the size of the images. We applied the same clipping operation to the benchmark in the test set as we did with the training set (mentationed in Appendix A.2.1), ensuring that the size of each initial pin image and its image of routing result ground truth is 64x64.

---

### Official Review · Reviewer_qcjE · 2025-03-14

**Overall Recommendation:** 4

**Summary:**

This paper introduces DSBRouter, a novel global routing (GR) solver leveraging the Diffusion Schrödinger Bridge (DSB) model. The authors aim to address the challenge of ensuring routing connectivity in network prediction results, a persistent issue in learning-based GR methods. DSBRouter learns both forward and backward mappings between initial pins and routing results, and incorporates an evaluation-based sampling scheme to enhance routing predictions. The results demonstrate state-of-the-art performance in overflow reduction on public benchmarks.

**Claims And Evidence:**

The primary claim that DSBRouter achieves state-of-the-art (SOTA) performance in overflow reduction is generally supported by the evidence provided.

**Essential References Not Discussed:**

No.

**Experimental Designs Or Analyses:**

- Ablation Study: The ablation study is designed to assess the contribution of the EG and NN modules.  The experimental setup is clear, and the results provide some insight into the importance of these components. However, as mentioned earlier, a more detailed analysis would be beneficial.

- Influence of Inferencing Steps: The experiments on the influence of inferencing steps are well-designed.  They systematically vary the number of steps and analyze the impact on performance metrics.

- Comparison with Baselines: The comparison with baseline methods is generally sound. The authors include both classical and ML-based routers in their comparison, providing a comprehensive evaluation of DSBRouter's performance.

**Methods And Evaluation Criteria:**

The proposed DSBRouter method is well-motivated. The use of the Diffusion Schrödinger Bridge model is novel in the context of global routing. The evaluation criteria is comprehensive.

**Other Comments Or Suggestions:**

No.

**Other Strengths And Weaknesses:**

No.

**Questions For Authors:**

The paper acknowledges that DSBRouter has a longer generation time. What specific strategies are the authors considering for future work to address this limitation, and what is the potential for these strategies to significantly improve the runtime?

The training data is generated using NthuRoute. How might the choice of training data generation method affect the generalizability of DSBRouter to different routing scenarios or design styles?

**Relation To Broader Scientific Literature:**

- Diffusion Models and Schrödinger Bridges: The paper builds upon the literature on diffusion models and Schrödinger Bridge models.  It extends these techniques to the problem of global routing. The authors also highlight the connection between SGMs and DSB.

- Global Routing Methods: The paper thoroughly reviews traditional and learning-based approaches to global routing.  It identifies the limitations of existing methods, such as the lack of connectivity guarantees and the reliance on post-processing.  DSBRouter is presented as a solution that addresses these limitations by providing an end-to-end approach that ensures connectivity.

- Objective Guidance: The use of objective guidance in the inference phase is inspired by techniques from score-based generative models.  The authors adapt these techniques to the DSB framework to improve the quality of the generated routes.

**Theoretical Claims:**

No.

---

> ### Author Rebuttal · Authors · 2025-04-01
>
> We would like to express our sincere gratitude for thoroughly evaluating our paper and providing insightful and valuable feedback. We are genuinely committed to addressing your concerns and respond to your specific comments below.
>
> > **Question 1: The paper acknowledges that DSBRouter has a longer generation time. What specific strategies are the authors considering for future work to address this limitation, and what is the potential for these strategies to significantly improve the runtime ?**
>
> Thanks for your valuable comment. The DSB in our proposed DSBRouter, like the standard [DDPM](https://proceedings.neurips.cc/paper/2020/hash/4c5bcfec8584af0d967f1ab10179ca4b-Abstract.html) and [SGM](https://proceedings.neurips.cc/paper/2019/hash/3001ef257407d5a371a96dcd947c7d93-Abstract.html?ref=https://githubhelp.com), is based on the diffusion and denoising processes, and therefore, efficiency concerns are also present. To address the efficiency issues, we consider the following two approaches:
>
> 1) adopt the techniques of the [Consistency Model](https://openreview.net/forum?id=FmqFfMTNnv) to reduce the sampling steps.
>
> 2) Recomputing the optimization objective every few steps during the inference process, instead of at each step, which simplifies the computational of evaluation-based guidance.
>
> While we are still implementing the two improvements, we estimate that they could reduce DSBRouter’s runtime by 60%-70%. Based on the 'Ablation Study' in Table 5, we believe runtime is closely tied to the number of inference steps. With a well-trained consistency model, we expect to compute the final route using only 1/5 of the original inference steps. Additionally, Strategy 2) reduces the complexity of optimizing the computational target, further cutting the runtime by ~60%. Final results will be included in our later rebuttal and final version.
>
> > **Question 2: The training data is generated using NthuRoute. How might the choice of training data generation method affect the generalizability of DSBRouter to different routing scenarios or design styles?**
>
> Thanks for your insightful comment. Regarding your question, we have the following conclusions:
>
> **Conclusion 1**: *The bias of the supervised training set towards specific metrics (OF and WL) does indeed affect the routing generalization performance of DSBRouter.*
>
> In the paper, the optimization objective of DSBRouter focuses on optimizing the OF metric. Therefore, we use the [Nthurouter](https://ieeexplore.ieee.org/document/4681595/) results on the datasets of [ISPD07](https://www.ispd.cc/contests/07/contest.html) including bigblue4, newblue4, newblue5, newblue6, and newblue7 as the supervised training set, as Nthurouter achieves SOTA performance on OF for these datasets compared to other methods proposed in the same year. The dataset newblue3, which has the highest number of pins, was excluded because its routing results performed poorly in terms of OF metrics. On the other hand, [Hubrouter](https://proceedings.neurips.cc/paper_files/paper/2023/hash/f7f98663c516fceb582354ee2d9d274d-Abstract-Conference.html) focuses on optimizing the WL metric and uses routing results from [NCTU-GR](https://ieeexplore.ieee.org/abstract/document/5703167), which are more tailored to optimizing the WL metric, as its supervised training set. The experimental results show that Hubrouter achieves SOTA performance on WL but lags behind on OF. To verify Conclusion 1, we replicated Hubrouter's training set and trained DSBRouter-NCTU using this set. We report the routing results of DSBRouter trained with two different training sets (other settings remain the same) on the ibm01 and ibm05 dataset in Table 1:
>
> Table 1. Comparision of DSBRouter-NTHU and DSBROUTER-NCTU on two different routing scenarios.
> | |ibm01-wirelength|ibm01-overflow|ibm05-wirelength|ibm05-overflow|
> | :-: | :-: | :-: | :-: | :-: |
> |DSBRouter-NTHU|61435|1430|420464|0|
> |DSBRouter-NCTU|86776|10256|1312353|115|
>
> To address the current issue of DSBRouter's insufficient generalization, we have recently attempted to improve the optimization objective proposed in Section 3.3 of the paper with a hyperparameter $\tau$. Where $\mathbb{S}(\mathcal{P}(\bar{\mathbf{x}_{k}}))$ turns to $\tau * E^o + (1-\tau) * E^w$. The goal is to enable DSBRouter to optimize a balance between OF and WL during inference. Due to space constraints, we only report the performance of DSBRouter on the ISPD98 cases in Table 2 for the cases where \($\tau = 0.3$ and $\tau = 0$\). (Experiments show that different datasets exhibit varying sensitivities to the hyperparameters $\tau$.) We will report the full results in future work.
>
> Table 2. Routing Results of DSBRouter on ISPD98 with modified objective.
> | |ibm01-wl/of |ibm02-wl/of|ibm03-wl/of|ibm04-wl/of|
> | :-: | :-: | :-: | :-: | :-: |
> |$\tau=0.3$|65665/1479|213588/0 | 165930/0|176945/0|
> |$\tau=0$|65632/1390|208405/0|165168/0|175869/0|

---

### Official Review · Reviewer_HZXX · 2025-03-14

**Overall Recommendation:** 4

**Summary:**

This paper introduces DSBRouter, an end-to-end neural global routing solver based on the Diffusion Schrödinger Bridge (DSB) model. Traditional learning-based approaches to global routing (GR) often require post-processing heuristics or reinforcement learning to enforce connectivity, leading to inefficiencies. In contrast, DSBRouter directly learns a bi-directional mapping between initial pins and final routing solutions, ensuring connectivity without the need for a second-stage correction. DSBRouter leverages a novel evaluation-based guidance mechanism to optimize routing outputs based on overflow minimization and connectivity constraints. Extensive experiments show that DSBRouter achieves SOTA.

**Claims And Evidence:**

Claim: The proposed DSBRouter achieves SOTA performance.
Evidence: Table 2 & 3

**Essential References Not Discussed:**

No.

**Experimental Designs Or Analyses:**

I think the experimental designs are not problematic.

**Methods And Evaluation Criteria:**

Methodology is okay. Benchmarks are canonical ones (ISPD07/98).

**Other Comments Or Suggestions:**

The formulae are way-too crowded in page 5 and 6. I think it might be better to consolidate them into simpler forms.

**Other Strengths And Weaknesses:**

Strengths:

Novelty. This is the first to introduce DSB to routing.

Weaknesses:

Efficiency is a big concern for the proposed method. (I would like to thank the authors' honesty for reporting this problem.)

**Questions For Authors:**

While I understand the constraint of page limits, the solution to faster inference in "limitations" is somehow vague. Could you go into details on how to improve the efficiency of the proposed method.

**Relation To Broader Scientific Literature:**

Unlike previous routing works, this paper is the first to introduce Diffusion Schrodinger Bridge to routing.

**Theoretical Claims:**

I briefly check the proof, but I could not ensure each and every detail is 100% correct.

---

> ### Author Rebuttal · Authors · 2025-04-01
>
> We would like to express our sincere gratitude for thoroughly evaluating our paper and providing valuable and constructive feedback. We are genuinely committed to addressing your concerns and respond to your specific comments below.
>
> > **Weakness: Efficiency is a big concern for the proposed method. (I would like to thank the authors' honesty for reporting this problem.) Could you go into details on how to improve the efficiency of the proposed method.**
>
> Thanks for your valuable comment. The DSB in our proposed DSBRouter, like the standard [DDPM](https://proceedings.neurips.cc/paper/2020/hash/4c5bcfec8584af0d967f1ab10179ca4b-Abstract.html) and [SGM](https://proceedings.neurips.cc/paper/2019/hash/3001ef257407d5a371a96dcd947c7d93-Abstract.html?ref=https://githubhelp.com), is based on the diffusion and denoising processes, and therefore, efficiency concerns are also present. To address the efficiency issues, we consider the following two approaches:
>
> 1) Adopt the techniques of the [Consistency Model](https://openreview.net/forum?id=FmqFfMTNnv) to reduce the sampling steps.
>
> 2) Recomputing the optimization objective every few steps during the inference process, instead of at each step, which simplifies the computational of evaluation-based guidance.
>
> We are working hard on implementing the above two strategies and the experiments which take days to finish. We will add the results in our later rebuttal and final version.
>
> > **Suggestion: The formulae are way-too crowded in page 5 and 6. I think it might be better to consolidate them into simpler forms.**
>
> Thanks for your insightful comment. We have revised the paper. The modified sections will be presented in our final version.
>
> > **Question: While I understand the constraint of page limits, the solution to faster inference in "limitations" is somehow vague.**
>
> Thanks for your comment. We have revised the statement in the 'limitations' section regarding the difficulty in achieving accelerated inference as follows: "Due to the nonmonotonically noise injection strategy and meaningful $p_r, p_s$ in the DSB-based sampling process, accelerated sampling techniques such as DDIM in DPM cannot be directly applied."
>
> The reasons why acceleration techniques similar to [DDIM](https://arxiv.org/abs/2010.02502) are difficult to implement in DSB are: In the diffusion process of DDIM, it is assumed that the noise $\epsilon_t = \sqrt{1-\bar{a}_t}\bar{z}_t$ intensity added at each time step monotonically increases as following:
> $$
> \begin{equation*}
> x_t=\sqrt{\bar{a}_t}x_0 + \sqrt{1-\bar{a}_t}\bar{z}_t \quad \bar{z}_t \sim \mathcal{N}(0,\mathbf{I}), \tag{1}
> \end{equation*}
> $$
> where $\bar{a}_t$ is a parameter that decreases with the increase of time $t$ and after $T$ steps of noise addition, $x_T$ will converge to a Gaussian distribution $\mathcal{N}(x_T,\sqrt{\bar{a}_T}x_0,(1-\bar{a}_T)\mathbf{I})$. During the denoising (sampling) process of DDIM, the neural network predicts the noise $\epsilon_t$ and DDIM have the following hypothesis:
>
> $$
> \begin{equation*}
> P(x_{t-1}|x_t,x_0) \sim \mathcal{N}(kx_0+mx_t,\sigma^2), \tag{2}
> \end{equation*}
> $$
>
> However, in DSB, since both $p_s$ ($x_0$) and $p_r$ ($x_T$) represent meaningful data distributions, the noise intensity cannot monotonically change during either the forward or backward processes. Thus, a symmetric noise scheduling scheme $\gamma_t$, as shown in Appendix A.2.1, is used. Additionally, supposing that the backbone of DSBRouter also predicts noise $\gamma_t$, predict this nonmonotonically changing target is intuitively more difficult, and since we employ IPF iteration to optimize DSB, if we were to sample using this predicted noise, the model’s training would become harder to converge. Besides, in the transition of $p_s \mapsto p_r$ (or $p_r \mapsto p_s$) in DSB, there is no hypothesis in equation (2). Therefore, techniques like DDIM for accelerated sampling can not directly be used to implement in the sampling process of DSB. Besides, the [DSB method](https://arxiv.org/abs/2403.14623) we follow did not use acclearated samping techniques, indicating that the use of acceleration sampling techniques in DSB still needs theoretical and practical verification.

---

### Decision · Program_Chairs · 2025-05-01

**Decision:**

Accept (poster)

**Comment:**

The paper proposed DSBRouter, an end-to-end neural global routing solver using Diffusion Schrödinger Bridge (DSB). The method shows good performance in real-world problems and all reviewers are positive.